# Keep Your Boundaries: From Finite Elements to Simplicial Convolution

## Abstract

Diffusion on graphs can be viewed from a perspective of partial differential equations (PDEs), while Graph Neural Networks (GNNs) can be interpreted as a discrete counterpart of diffusion PDEs. As such, many emerging GNNs have been inspired by PDE solvers from numerical analysis. However, most PDE-assisted GNNs employ finite-difference methods, which essentially consider only the function discretization, evaluated on the grid nodes. We propose to bring the ideas of finite element methods (FEMs) from numerical PDEs to GNNs, allowing us not only to define the function on the local region (simplices) but to enforce the appropriate continuity constraints on the neighboring simplices. We develop a novel Simplicial Element Network (SEN) framework with the dual simplicial convolution encoder, based on a *specially-designed interior boundary matrix* to enhance the graph representation learning. We discuss theoretical underpinnings of SEN, using the notion of combinatorial Laplacians on simplices and discrete Hodge theory. Our experiments indicate that SEN outperforms 20+ state-of-the-art baseline methods on 10+ benchmark datasets on 5 tasks: protein sequence recovery, link prediction, air quality forecasting, trajectory prediction, and fluid flow reconstruction.

## 1 Introduction

One of the newest directions in machine learning focuses on the nexus of graph neural networks (GNNs) and partial differential equations (PDEs) (52). This trend is inherently a two-way highway. Indeed, one path along this route (and arguably one of the primary headlines in scientific computing) aims to improve numerical PDE solvers by combining them with GNNs. Such GNN-enabled PDE solvers allow us to deliver solutions for arbitrary locations and handle unstructured grids and irregular time intervals of complex dynamical systems (12; 32; 44). Reversely, the message passing mechanism (i.e., the backbone of GNNs) is essentially some form of diffusion. As such, GNNs may be viewed as a discretized version of PDEs, or neural diffusion PDEs (8), and many recently emerging GNN architectures are inherently inspired by the PDE-based approaches from computational mathematics and scientific computing (20; 57).

In turn, in numerical analysis, the primary discrete representations of a function in PDE can be generally classified into node-values, element-basis, and spectral basis (23). The first two classes are from finite-difference methods (FDMs) and finite element methods (FEMs), respectively. More specifically, in FDMs, the region where the function is defined is discretized into a grid, and the function is discretized and evaluated on the nodes (of the grid). Stated differently, a continuous function is now represented by a set of node values. In FEMs, the computation domain is discretized into meshes, and the function is represented by a set of basis functions. Each of the base function is defined on a local region composed of several triangles (or tetrahedrons). More importantly, the basis functions will satisfy certain constraints, for instance, they are continuous on the boundary of two adjacent regions. (Indeed, intuitively, weather on the border of Germany and France is the same regardless of whether you are on the German or French side.) Figure 1 shows the two types of function representations in scientific computing. Turning back to graph learning, the existing PDE-based approaches for GNNs largely focus on FDMs. The closest philosophy to FEM is arguably exhibited by simplicial neural networks (SNNs) which model diffusion across higher-order graph substructures described by simplices. Such

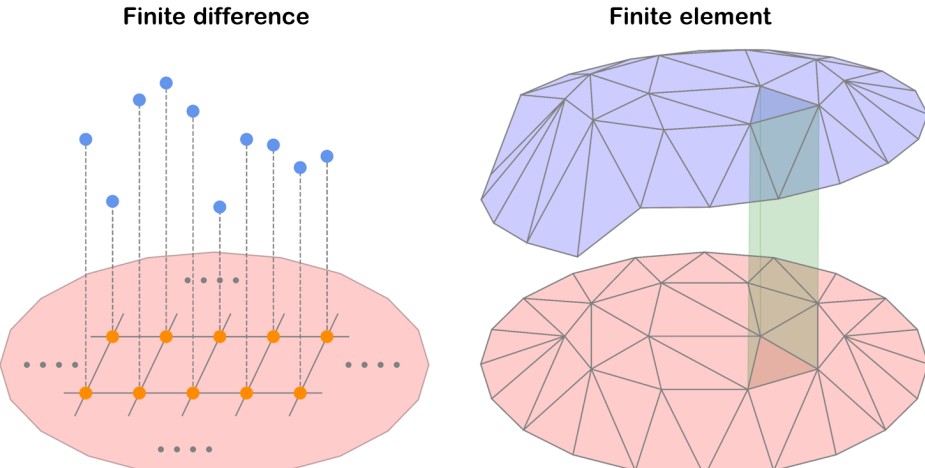

Figure 1: Function representations: finite differences and finite elements. In finite element models, the basis functions (i.e., the piece-wise function in blue color) are continuous on the edges of the adjacent triangles in the interior region. This is the "continuity" constraint considered in our SEN model.

simplices represent a natural analogue of local regions in FEM. However, SNNs do not impose any restriction on the *continuity of function on* adjacent regions, thereby disregarding this important information.

We address this fundamental gap by bridging the emerging directions of simplicial diffusion and PDE-inspired GNN architectures with the finite element methodology from numerical analysis. By invoking combinatorial Laplacians on simplices and elements of discrete Hodge theory, we formulate a new simplicial element network (SEN), which then translates the idea of FEM to graph structures and allows us to systematically enforce *continuity* conditions across graph (sub)structures of varying dimensions. We discuss theoretical underpinnings behind SEN and test its versatility in a broad range of domain applications. In particular, to illustrate the critical role of *continuity* conditions, we apply SEN to forecasting of air quality due to the recent wildfires in Quebec (clearly, air quality is on the border of Quebec and New York state ought to be the same!), trajectory prediction in synthetic flows, and fluid flow reconstruction, as well as protein sequence recovery and link prediction. The key contributions of our approach can be summarized as follows:

- We introduce the concepts of finite element methods from numerical analysis of PDEs to graph neural diffusion, thereby explicitly addressing the physics-inspired notion of *continuity* constraints with a *specially-designed interior boundary matrix*.

- We develop a new simplicial element network (SEN) which enables us both to holistically describe diffusion across higher-order graph (sub)structures of varying dimensions and to systematically address the *continuity* conditions among neighboring (sub)structures.

- By invoking combinatorial Laplacians on simplices and elements of discrete Hodge theory, we discuss theoretical underpinnings of SEN.

- We validate SEN utility in application to 5 tasks: inverse protein folding, link prediction, air quality forecasting, trajectory prediction, and fluid flow reconstruction. Our extensive experiments indicate that SEN outperforms 20+ state-of-the-art baselines on all 10+ datasets.

## 2 Related Works on Simplicial Complex Learning

Modeling higher-order interactions on graphs is an active and rapidly evolving research direction in graph representation learning (4; 48; 58; 41). Although the role of higher-order structures in graph learning has been

documented for many years (1; 34), encompassing complex phenomena ranging from disease transmission to biodiversity to money laundering, their systematic incorporation into graph deep learning through simplicial representations emerged only in 2020. As demonstrated by (17; 47; 5; 11; 39; 19), simplicial representations can improve graph learning performance across a wide range of tasks, especially under heterogeneous settings. The related ideas on triangle-mesh-based networks have been also explored in the graphics community (53; 50), as well as via the lens of Message-Passing Graph Neural Networks (MPNNs) (18; 67), which expressivity has recently been studied by (45; 67; 70). Recent approaches based on sheaf neural networks further advance these ideas by designing learnable linear maps to describe how information propagates across the graph (29; 2; 14; 27; 46) . However, neither of these simplicial methods account for the important *continuity* conditions in the diffusion process over graph. In particular, motivated by FEM for PDE solutions, we introduce a general definition of the boundary regions and continuity conditions which allows us to systematically treat a wide range of realistic scenarios, including multiple (overlapping) simplices.

## 3   Background and Motivation

Consider a graph $\mathcal{G} = \{\mathcal{V}, \mathcal{E}\}$, where $\mathcal{V}$ is a node set with cardinality $|\mathcal{V}|$ of $N$ and $\mathcal{E} \subseteq \mathcal{V} \times \mathcal{V}$ is an edge set. Let $N \times N$-matrix $\boldsymbol{A}$ with entries $a_{uv} \neq 0$ for any $e_{uv} \in \mathcal{E}$ and $a_{uv} = 0$, otherwise, be an adjacency matrix of $\mathcal{G}$. Let $\mathbf{f}$ be a real function specified on nodes of $\mathcal{G}$, i.e. $\mathbf{f} : \mathcal{V} \to \mathbb{R}^d$ such that for each $u_i \in \mathcal{V}$ and $\mathbf{f}(u_i) = \mathbf{x}_{u_i}$, where $d$ is the number of features and $\boldsymbol{x}_{u_i}$ are node features of $u_i$.

**Preliminaries on Simplicial Complexes and Their (Co)-Boundaries** A simplicial complex can be viewed as a generalization of graphs into their higher-dimensional counterpart (see Appendix A for further details). As such, geometrically nodes can be interpreted as 0-simplices, edges as 1-simplices, triangles as 2-simplices, tetrahedrons as 3-simplices etc. Since our primary focus is on enforcing the natural requirement that the feature function takes the same value along the boundary of the two neighboring higher-order graph (sub)structures, we need to define the notion of "neighbors".

**Definition 3.1.** Let $\mathcal{K}$ be a finite-dimensional simplicial complex and $\sigma^1$ and $\sigma^2$ be two $p$-dimensional simplices in $\mathcal{K}$, $0 \leq p \leq dim(\mathcal{K})$. Then

(i)  $\sigma^1$ and $\sigma^2$ are *lower adjacent neighbors*, denoted as $\sigma^1 \smile \sigma^2$, if $\sigma^1$ and $\sigma^2$ both contain some nonempty $(p-1)$-simplex $\mu$ as a face. I.e., there exists a non-empty $\mu$ in $\mathcal{K}$ such that $\mu < \sigma^1$ and $\mu < \sigma^2$. Then such $\mu$ is called a *common lower simplex* of $\sigma^1$ and $\sigma^2$.

(ii)  $\sigma^1$ and $\sigma^2$ are *upper adjacent neighbors*, denoted as $\sigma^1 \frown \sigma^2$, if $\sigma^1$ and $\sigma^2$ are both faces of some $(p+1)$-simplex $\gamma$, i.e. there exists a $\gamma$ in $\mathcal{K}$ such that $\gamma > \sigma^1$ and $\mu > \sigma^2$, and $\gamma$ is called their *common upper simplex*.

To describe the diffusion among $p$-simplices of $\mathcal{G}$ using linear maps and to enforce the associated *continuity* conditions for neighboring simplices, we form a real-valued vector space $C_p$ which is endowed with a basis from the oriented $p$-simplices. Elements of $C_p$ are called *p-chains*. (See Appendix A for more details on $C_p$ and its relations to boundaries.)

**Motivation:** Given the recent ideas on message propagation across simplices and the associated simplicial convolution, it would be natural to extend the node-wise feature function $\mathbf{f}$ to simplices of $\mathcal{G}$. That is, $\mathbf{f}$ now takes values on elements of a simplicial complex $\mathcal{K}$ over $\mathcal{G}$, i.e., $\mathbf{f} : \mathcal{K} \to \mathbb{R}^d$. Learning $\mathbf{f}$ may be associated with a wide range of tasks, from node classification (i.e. function on nodes) to link prediction (i.e., function on edges) to subgraph prediction (i.e., function on subgraphs). Hence, the next immediate question is: **How can we better learn function f?** We aim to address it through the lens of simplicial convolution on graphs and finite element methods for PDE solutions.

**Higher-Level Overview of Function Learning: Message and Aggregation from Nodes to Simplices** Message module and aggregation module are the most fundamental mechanisms for function learning in GNNs. The message module (MSG) computes a message $\mathbf{m}^{(l)}(u)$ for node $u$ at $l$-th layer

$$\mathbf{m}^{(l)}(u) = \text{MSG}^{(l)}(\mathbf{f}^{(l-1)}(v), v \in \{\mathbb{N}(u)\}).$$

Here $\mathbf{f}$ is the node feature and $\{\mathbb{N}(v)\}$ is a set of neighbors of $v$. Then, the aggregation module (AGG) aggregates messages from "neighbors"

$$\mathbf{f}^{(l)}(v) = \text{AGG}^{(l)}(\{\mathbf{m}^{(l)}(u), u \in \mathbb{N}(v)\}, \mathbf{m}^l(v)).$$

The definition of "neighbors" is very natural for graphs, i.e., nodes form neighbors if there is a common edge. If we now turn to simplicial learning, feature function $\mathbf{f}$ is to be defined on $p$-simplex $\sigma^p$ (we omit superscript $p$ for the sake of notation). The message module then takes a form

$$\mathbf{m}^{(l)}(\sigma) = \text{MSG}^{(l)}(\mathbf{f}^{(l-1)}(\sigma')), \sigma' \cap \sigma \neq \emptyset,$$

where $\sigma'$ is a $p$-simplex neighboring $\sigma$. The relation for "neighbors" here can be defined as either upper adjacency (i.e. $\sigma' \frown \sigma$, which means two simplices share a coface) or lower adjacency (i.e. $\sigma' \smile \sigma'$, which means that they share a face). (Other neighboring relations can also be considered.)

Module AGG then aggregates messages from "neighbors", e.g., lower adjacent simplices:

$$\mathbf{f}^{(l)}(\sigma) = \text{AGG}^{(l)}(\{\mathbf{m}^{(l)}(\sigma'), \sigma' \cap \sigma \neq \emptyset\}, \mathbf{m}^{(l)}(\sigma)).$$

The aggregation module can also consider the messages from higher or lower dimensional simplices. For instance, the messages from faces or cofaces can be used for aggregation in simplicial learning.

To the best of our knowledge, **no existing SNNs or GNNs consider the natural condition that function f shall satisfy certain equivalence properties on the boundary shared by two simplices**.

To preserve this important "consistency" property, we can enforce an explicit *continuity* condition on their boundary. The most general one is to ensure certain "equivalence" relations. For instance, we can define a boundary function

$$G : \mathbf{f}^{(l)}(\sigma) \to \mathbf{g}^{(l)}(\sigma)|_{\mu}, \mu < \sigma,$$

where $\mu$ is the face of $\sigma$, and $\mathbf{g}^{(l)}(\sigma)|_{\mu}$ is the boundary function (vector) for $\mathbf{f}^{(l)}(\sigma)$. If two simplices share a face, e.g., $\mu = \sigma \bigcap \sigma'$, the *continuity* constraint can be

$$\mathbf{g}^{(l)}(\sigma)|_{\mu} = \mathbf{g}^{(l)}(\sigma')|_{\mu}.$$

From the finite element point of view, that is to say the base functions $\mathbf{f}^{(l)}(\sigma)$ and $\mathbf{f}^{(l)}(\sigma')$ share some common values on their boundary, i.e., they are continuous functions on their boundary.

**Remark** Note that we can also set up different *continuity* conditions, for instance, we can require the derivatives of the functions to be continuous at their boundary. Indeed, as in our illustrative example on air quality forecasting due to Quebec wildfires (see the right panel of Table 1), air quality on the border of Quebec and New York State ought to be the same regardless of whether you are on the Canadian or U.S. side. In applications, such as molecular data analysis, the functions can be related to physical properties, such as energy, electro-static potential, and potential field.

## 4 Simplicial Element Network

Our goal is to bridge the finite element methods (FEMs) for numerical PDE solvers with the GNN function representations. That is, we target not only message passing mechanisms through higher order interactions among graph (sub)structures, as SNNs do, but importantly we aim to systematically account for *continuity* constraints among them. In particular, we ensure the intuitive requirement that whenever two graph (sub)structures, or coarser-resolution elements described by simplicial complexes, share the same boundary, the function on the boundary shall be the same for both elements (41). We approach this goal by introducing the boundary matrices and the associated combinatorial Laplacians on simplicial complexes (often referred to as Hodge Laplacians). Such combinatorial Laplacians serve as extensions of the conventional graph Laplacian to a higher-order (sub)structures, allowing us to model both the message propagation across such graph (sub)structures and to ensure the important continuity conditions, on par with the FEM ideas for PDE solutions.

**Definition 4.1.** Let $\mathcal{K}$ be a finite oriented simplicial complex, $p \in \mathbb{Z}^+$, $p \leq dim(\mathcal{K})$. Let $\mathcal{A}$ be a set of $p$-dimensional simplices $\sigma$ of $\mathcal{K}$ (i.e., $\sigma^p$) and $\mathcal{B}$ be a set of $(p-1)$-dimensional simplices $\sigma$ of $\mathcal{K}$ (i.e., $\sigma^{p-1}$). Then we define an *incidence* matrix $\mathbf{B}$

$$\mathbf{B}_p(i,j) = \begin{cases} 1, & \text{if } \sigma_i^{p-1} < \sigma_j^p, \sigma_i^{p-1} \sim \sigma_j^p \\ -1, & \text{if } \sigma_i^{p-1} < \sigma_j^p, \sigma_i^{p-1} \nsim \sigma_j^p \\ 0, & \text{otherwise,} \end{cases}$$

Matrix $\mathbf{B}$ records which $(p-1)$-simplices $\sigma^{p-1}$ serve as faces of $p$-simplices $\sigma^p$, essentially conveying information on the lower boundary of $\sigma^p$, and, hence, is also referred to as $p$-th *boundary operator* $C_p \to C_{p-1}$. For $p = 1$, $\mathbf{B}_p$ coincides with the familiar node-edge incidence matrix for graphs, and for higher $p$, $\mathbf{B}$ can be used to compute homology of $\mathcal{K}$, that is, the number of $p$-dimensional voids and connected components of a topological space.

**Definition 4.2.** Armed with $\mathbf{B}_p$, similarly to the lower dimensional case, we can define the corresponding lower and upper combinatorial Laplacian matrices $\mathbf{L}_p^{\text{down}} = \mathbf{B}_p^\top \mathbf{B}_p$ and $\mathbf{L}_p^{\text{up}} \mathbf{L}_p^{\text{up}} = \mathbf{B}_{p+1} \mathbf{B}_{p+1}^\top$, respectively. That is, $\mathbf{L}_p^{\text{down}} : C_p \to C_p$ and $\mathbf{L}_p^{\text{up}} : C_p \to C_p$. (See Appendix A for further details.).

**Proposition 4.3.** *Linear operators $\mathbf{L}_p^{down}$ and $\mathbf{L}_p^{upper}$ are symmetric and semi-positive definite. (Proof is in Appendix B.)*

Finally, using the boundary operator $\mathbf{B}_p$ and $\mathbf{B}_{p+1}$, we can also define the combinatorial Laplacian $\mathbf{L}_p$, generalizing the notion of the conventional graph Laplacian to simplices and allowing us to describe diffusion through higher-order graph (sub)structures, e.g., from edges to edges via triangles:

$$\mathbf{L}_p = \mathbf{L}_p^{\text{down}} + \mathbf{L}_p^{\text{up}} = \mathbf{B}_p^\top \mathbf{B}_p + \mathbf{B}_{p+1} \mathbf{B}_{p+1}^\top.$$

Note that $\mathbf{L}_p$ is also often referred to discrete Hodge-Laplacian on graphs (37).

**Proposition 4.4.** *The three combinatorial Laplacians, i.e., $\mathbf{L}_p^{down}$, $\mathbf{L}_p^{up}$, $\mathbf{L}_p$ correspond to three different types of simplex message passing modules.*

Indeed, from the definition of the combinatorial Laplacians on simplices, it can be seen that the pattern of their off-diagonal values are very different. Specifically, for $\mathbf{L}_p^{\text{up}}$, two simplices are "neighbors" only if they are upper adjacent. For $\mathbf{L}_p^{\text{down}}$, two simplices are "neighbors" only if they are lower adjacent. For $\mathbf{L}$, two simplices are "neighbors" only if they are lower adjacent but not upper adjacent.

However, this scheme may not necessarily *highlight* and learn certain $p$-dimensional simplices which receive messages from multiple $(p + 1)$-dimensional simplices, e.g., cofaces. For instance, when $p = 1$ (i.e., 1-dimensional simplices/edges), edges are shared by more than one triangle. To address this issue, we propose a novel simplicial convolutional encoder with special *continuity* constraints for function in "interior" regions, and our encoder focuses on modeling $p$-dimensional simplices which receive message information from multiple higher dimensional simplices (i.e., next higher rank).

**Proposition 4.5.** *Let $\tilde{\mathcal{K}}$ be a finite dimensional complex without orientation and let $p$ be an integer such that $0 \leq p \leq dim(\tilde{\mathcal{K}})$. Then $\tilde{\mathbf{L}}_p^{down} = \tilde{\mathcal{B}}_p^\top \tilde{\mathcal{B}}_p$ is diagonally dominant, symmetric and $\tilde{\mathbf{L}}_p^{down} \geq 0$. (Proof is in Appendix B.)*

We now turn to the interior boundary matrix.

**Definition 4.6.** Interior boundary matrix $\tilde{\mathbf{B}}_{p+1}$ is given by

$$\tilde{\mathbf{B}}_{p+1}(i,j) = \begin{cases} \mathbf{B}_p(i,j), & \text{if } \sum_j |\mathbf{B}_p(i,j)| > 1 \\ 0, & \text{otherwise,} \end{cases}$$

and the corresponding *interior* upper combinatorial Laplacian $\tilde{\mathbf{L}}_p^{\text{up}} = \tilde{\mathbf{B}}_{p+1} \tilde{\mathbf{B}}_{p+1}^\top$. Note that $\tilde{\mathbf{L}}_p^{\text{up}}$ is a linear operator that is defined only on simplices that have at least two cofaces, i.e., simplices are within the interior region (not on the boundary of the domain). Specifically, the entries of $\tilde{\mathbf{L}}_p^{\text{up}}$ are given by

$$\tilde{\mathbf{L}}_p^{\text{up}}(i,j) = \begin{cases} \mathbf{L}_p^{\text{up}}(i,j), & deg_U(\sigma_i^p) > 1 \\ 0, & \text{otherwise.} \end{cases}$$

**Remark** The *interior* upper combinatorial Laplacian matrix $\tilde{\mathbf{L}}_p^{\text{up}}$ is symmetric, and the message passing module based on $\tilde{\mathbf{L}}_p^{\text{up}}$ is different from all previous ones based on $\mathbf{L}_p^{\text{down}}$, $\mathbf{L}_p^{\text{up}}$, and $\mathbf{L}_p$.

Figure 2 shows an example of normal boundary matrix $\mathbf{B}_2$ and *interior* boundary matrix $\tilde{\mathbf{B}}_2$ for the same simplicial complex. In our $\tilde{\mathbf{B}}_2$, only the edge (1-simplex) that has at least two triangle cofaces will be considered.

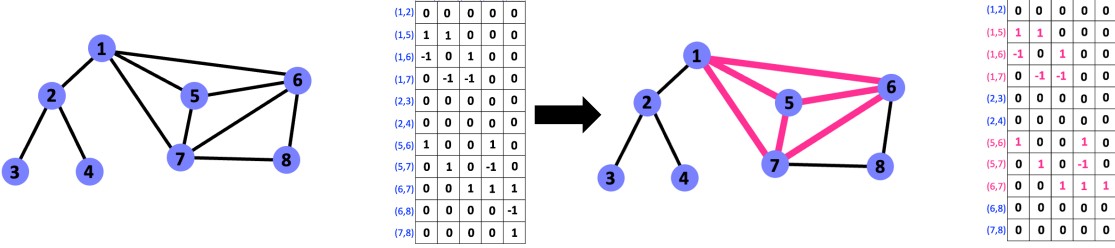

Figure 2: Comparison between the boundary matrix $\mathbf{B}_2$ (*left*) and interior boundary matrix $\tilde{\mathbf{B}}_2$ (*right*).

From the FEM point of view, these edges are within the "interior" region of the computational domain. Edges with zero or one coface form the "boundary" of the computational domain. In the "interior" region, base functions should satisfy continuity properties on the shared edges (boundary of two adjacent triangles). Hence, extra *continuity* conditions are enforced on these edges in our SEN model. Note that the difference lies on the edge that has only one coface, i.e., the edge is the boundary of only one triangle. For instance, edges $e_{68}$ and $e_{78}$ are located within only one triangle, and the corresponding terms in the interior boundary matrix are all zero. Topologically, all the edges (1-simplices) can be classified into three types, i.e., zero coface, one coface, and at least two cofaces. From the finite element point of view, edges with zero or one coface are the "boundary" of the computational domain, while edges with at least two cofaces are within the "interior" region of the computational domain and special continuity constraints for basis functions should be enforced on these edges.

## 5 Dual Simplicial Convolutional Encoder

Graphs such as proteins and chemical compounds often reveal intricate dependencies within their substructures, which are far more complex than what dyadic interactions or pairwise node relations can depict. For instance, protein-protein interactions are closely linked to protein essentiality, that is, their importance for cellular survival and function (61), suggesting that higher-order structural relationships play a crucial role and should be explicitly modeled. The discrete Hodge theory offers a systematic approach to address these higher-order polyadic interactions (37; 22). Specifically, the discrete Hodge theory extends the concept of the conventional combinatorial graph Laplacian, i.e., describing diffusion from one node to another via edges on graph $\mathcal{G}$ to encapsulate diffusion across higher-order substructures of $\mathcal{G}$. These substructures can be represented as $k$-simplices of $\mathcal{G}$. Consider, for example, a biomolecular system, where bond angles can only be defined between two atomic bonds and hence, three atoms should be involved simultaneously. A proper definition of dihedral angle requires four adjacent atoms at once. Both angles are of key importance in biomolecular structure and function characterization, and are the integral components in the force field of molecular dynamics. Hence, to model such biomolecular properties, we need to account both for higher-order structures and their associated *continuity* conditions. Our objective is to weave in the emerging concepts of simplicial convolution and introduce Hodge Laplacians at different dimensions to capture the varying dependencies between simplices, along with the *continuity* constraint. We design the higher-order graph convolutional encoder to (i) perform higher-order convolutions, (ii) encode multi-scale higher-order information, (iii) encode domain-specific edge information, and (iv) mix different dimensions in simplicial complexes and provide an efficient way to explore each dimension for graph representation learning. To address these goals, we propose a novel dual simplicial convolutional encoder that performs simplicial convolution operations over multi-dimensional simplices.

To extract the node embeddings from $p$-simplex, we propose Dual Simplicial Convolutional Encoder which contains two parts. The first part is based on the general simplex message passing model

$$\mathbf{Z}_{p,\mathcal{S}}^{(\ell+1)} = \sigma(\mathbf{L}_p \mathbf{Z}_{p,\mathcal{S}}^{(\ell)} \mathbf{\Theta}_{p,\mathcal{S}}^{(\ell)}). \tag{1}$$

Here $\mathbf{L}_p$ is the $p$-th combinatorial Laplacian, $\mathbf{\Theta}_{p,\mathcal{S}}^{(\ell)}$ stands for trainable weights, $\mathbf{Z}_{p,\mathcal{S}}^{(\ell)}$ and $\mathbf{Z}_{p,\mathcal{S}}^{(\ell+1)}$ are input and output activations of the $\ell$-th simplicial convolutional layer for the $p$-simplex. For $\mathbf{Z}_{p,\mathcal{S}}^{(0)}$, for instance, when $p = 1$, $\mathbf{Z}_{1,\mathcal{S}}^{(0)}$ represents the edge feature matrix, i.e., $\mathbf{z}_{uv,1,\mathcal{S}}^{(0)} = f(\mathbf{x}_u, \mathbf{x}_v)$ (which is the same as $\tilde{\mathbf{Z}}_{1,\mathcal{S}}^{(0)}$); when $k = 2$, $\mathbf{Z}_{2,\mathcal{S}}^{(0)}$ is the triangle feature matrix, i.e., $\mathbf{z}_{uvw,2,\mathcal{S}} = f(\mathbf{x}_u, \mathbf{x}_v, \mathbf{x}_w)$, where the $f(\cdot)$ is an aggregation functions (e.g., summation, maximum, and mean).

Then, after several message passing layers, we pass all the information to nodes. That is, after conducting the above simplicial convolution operations over $\mathcal{P}$ different simplices (i.e., Eq. 1), we convert each $p$-simplex embedding into the node-level representation via the boundary matrices and multi-layer perceptrons (MLPs)

$$\begin{aligned}
\bar{\mathbf{Z}}_{1,\mathcal{S}}^{(\ell+1)} &= \mathbf{B}_0(\mathrm{MLP}((\mathbf{Z}_{1,\mathcal{S}}^{(\ell+1)}))), \quad p = 1, \\
\bar{\mathbf{Z}}_{2,\mathcal{S}}^{(\ell+1)} &= \mathbf{B}_0(\mathrm{MLP}(\mathbf{B}_1(\mathrm{MLP}(\mathbf{Z}_{2,\mathcal{S}}^{(\ell+1)})))), \quad p = 2, \\
\bar{\mathbf{Z}}_{p,\mathcal{S}}^{(\ell+1)} &= \mathbf{B}_0(\mathrm{MLP}(\mathbf{B}_1(\mathrm{MLP}(\dots(\mathbf{B}_p(\mathrm{MLP}(\mathbf{Z}_{p,\mathcal{S}}^{(\ell+1)})))))))), \quad p > 2,
\end{aligned}$$

where $\mathbf{B}_p$ is the boundary matrix and here the MLP can avoid $\mathbf{B}_i \mathbf{B}_{i+1} = 0$.

In this way, we get $\mathcal{P}$ embeddings as $\{\bar{\mathbf{Z}}_{1,\mathcal{S}}^{(\ell+1)}, \dots, \bar{\mathbf{Z}}_{p,\mathcal{S}}^{(\ell+1)}, \dots, \bar{\mathbf{Z}}_{\mathcal{P},\mathcal{S}}^{(\ell+1)}, \tilde{\mathbf{Z}}_{1,\mathcal{S}}^{(\ell+1)}\}$, where $\mathcal{P} \geq 1$. This part can be viewed as the standard simplex message passing module. Now to enforce the *continuity* conditions, we design another special message-passing mechanism only on $(p-1)$-simplices shared by more than $\sigma_i^p$ that is applied iteratively in all layers:

$$\mathbf{H}^{(\ell+1)} = \mathbf{B}_0(\sigma(\tilde{\mathbf{L}}_1^{\mathrm{up}} \tilde{\mathbf{Z}}_{1,\mathcal{S}}^{(\ell)} \tilde{\mathbf{\Theta}}_{1,\mathcal{S}}^{(\ell)})) \tag{2}$$

and pass the information to nodes $\hat{\mathbf{H}}^{(\ell+1)} = \mathbf{B}_0 \mathbf{H}^{(\ell+1)}$. That is, $\hat{\mathbf{H}}^{(\ell+1)}$ is the output from the constrained 2-simplex convolutional encoder at $\ell$-th layer, resulting in $(\mathcal{P}+1)$ embeddings:

$$\{\bar{\mathbf{Z}}_{1,\mathcal{S}}^{(\ell+1)}, \dots, \bar{\mathbf{Z}}_{p,\mathcal{S}}^{(\ell+1)}, \dots, \bar{\mathbf{Z}}_{\mathcal{P},\mathcal{S}}^{(\ell+1)}, \tilde{\mathbf{Z}}_{1,\mathcal{S}}^{(\ell+1)}, \hat{\mathbf{H}}^{(\ell+1)}\}.$$

Note here that we apply the proposed *continuity* message passing operation for *all orders of the considered simplices and in all layers iteratively.* The key idea to have such a special *continuity* message-passing for shared-edges is to alleviate the gap between simplex vector (function) quantities, i.e., adding the *continuity constraint. More specifically, in finite element situations, we require the basis functions to be continuous on the shared edge of two adjacent triangles.* Note that how to define the boundary value of simplex vector/function is non-trivial. This can be done by using the boundary operator $\partial^*$, or we can define some special boundary operators, such as, a certain MLP structure. We do not explicitly specify the formula for boundary operator in SEN, instead we consider a special message passing module **only** on edges that are shared by triangles and use it as an approximation of *continuity* constraints. The general idea is that, in prevailing GNNs or SNNs, their *continuity* constraints are usually not satisfied, i.e., $\partial^* f_1 | \{v_2, v_3\}$ does not equal to $\partial^* f_2 | \{v_2, v_3\}$. Note that this difference only happens when the 2-simplices $(\{v_1, v_2, v_3\}, \{v_2, v_3, v_4\})$ share a common edge $(\{v_2, v_3\})$. Hence, we define a special *continuity* message-passing module on these edges (shared by at least by 2-simplices) to compensate or reduce the gap in their boundary values. Computationally, we use the $\tilde{\mathbf{L}}_p^{\mathrm{up}}$ matrix-based message-passing module. Note that this part is used to make sure the gap between boundaries values from 2-simplices is consistent. For example, in biomolecular studies many of the molecular properties, such as electro-density function and electro-static potential, are usually continuous functions. In this way, it is natural to require the learned latent features to preserve certain continuity over the boundaries.

Considering the node state can be correlated with one of them or even their combinations, we use the attention mechanism to focus on the importance of task relevant parts of the learned representations for

decision making, i.e., $(\alpha_1, \ldots, \alpha_p, \alpha_{\mathcal{P}+1}) = Att(\bar{\mathbf{Z}}_{1,\mathcal{S}}, \ldots, \bar{\mathbf{Z}}_{\mathcal{P},\mathcal{S}}, \tilde{\mathbf{Z}}_{1,\mathcal{S}})$. For the sake of simplicity, we omit the superscript $(\ell + 1)$. In practice, we compute the attention coefficient as:

$$\alpha_p = \frac{\exp\left(\Upsilon_{\text{Att}}\phi(\mathbf{\Phi}\bar{\mathbf{Z}}_{p,\mathcal{S}})\right)}{\sum_p^{\mathcal{P}} \exp\left(\Upsilon_{\text{Att}}\phi(\mathbf{\Phi}\bar{\mathbf{Z}}_{p,\mathcal{S}})\right) + \exp\left(\Upsilon_{\text{Att}}\phi(\mathbf{\Phi}\tilde{\mathbf{Z}}_{1,\mathcal{S}})\right)},$$

$$\alpha_{\mathcal{P}+1} = \frac{\exp\left(\Upsilon_{\text{Att}}\phi(\mathbf{\Phi}\tilde{\mathbf{Z}}_{1,\mathcal{S}})\right)}{\sum_p^{\mathcal{P}} \exp\left(\Upsilon_{\text{Att}}\phi(\mathbf{\Phi}\bar{\mathbf{Z}}_{p,\mathcal{S}})\right) + \exp\left(\Upsilon_{\text{Att}}\phi(\mathbf{\Phi}\tilde{\mathbf{Z}}_{1,\mathcal{S}})\right)},$$

where $\phi(\cdot)$ denotes a nonlinear activation function, $\Upsilon_{\text{Att}} \in \mathbb{R}^{1 \times d_{\text{out}}}$ is a linear transformation, $\mathbf{\Phi}$ is the trainable weight matrix, and the softmax function is used to normalize the attention vector. As a result, we obtain the final simplex embedding $\mathbf{Z}_{\mathcal{S}}$ by combining all embeddings

$$\mathbf{Z}_{\mathcal{S}} = \alpha_1 \times \bar{\mathbf{Z}}_{1,\mathcal{S}} \cdots + \alpha_{\mathcal{P}} \times \bar{\mathbf{Z}}_{\mathcal{P},\mathcal{S}} + \alpha_{\mathcal{P}+1} \times \tilde{\mathbf{Z}}_{1,\mathcal{S}}.$$

We then combine the embeddings from both conventional graph convolutional encoder and explainable simplicial convolutional encoder $\mathbf{Z} = \rho_{\mathcal{G}} \times \mathbf{Z}_{\mathcal{G}} + \rho_{\mathcal{S}} \times \mathbf{Z}_{\mathcal{S}}, \rho_{\mathcal{G}}, \rho_{\mathcal{S}} \in [0, 1]$, where $\rho_{\mathcal{G}}$ and $\rho_{\mathcal{S}}$ are hyperparameters which control the influence of different modules.

## 6 Experiments

### 6.1 Experimental Settings

We validate SEN on 5 tasks: protein sequence recovery, link prediction, air quality forecasting, trajectory prediction, and fluid flow reconstruction with respect to 20+ baseline models on 10+ benchmark datasets. The best results are in bold.

For **forecasting tasks**, we use wildfire datasets (54; 13), namely, PM2.5 concentrations from AirNow observations originating from the Quebec wildfire. We compare to the following popular models as baselines: (i) ResNet (30), (ii) PDE-Net 2.0 (38), (iii) GNN-PDE (32), and (iv) Simplicial Neural Networks-PDE (SNN-PDE) (13).

For **protein sequence recovery tasks**, we use three different real-world datasets CATH (protein class (C), architecture (A), topology (T) and homologous superfamily (H)) (40), TS50 and TS500 introduced by (43). We compare SEN with the powerful baselines, including StructGNN (33), GCA (56), GraDe-IF (66), and PiFold (25). PiFold is a protein inverse folding method that introduces a unique residue featurizer and employs stacked PiGNNs layers. The residue featurizer constructs more comprehensive features for each residue and introduces learnable virtual atoms to capture information overlooked by real atoms. The PiGNN considers feature dependencies at the node, edge, and global levels to learn from multi-scale residue interactions, eliminating the autoregressive decoder by stacking more PiGNN layers without sacrificing accuracy.

For **link prediction tasks**, we use three types of networks (i) citation and collaboration networks: Cora and PubMed (49), and large-scale **ogbl-collab** (31); (ii) social networks: (1) flight network: Airport (9), and (2) criminal networks: Sicilian Mafia Meetings and Phone Calls (7); (iii) disease propagation tree: Disease (9); (iv) the large-scale drug-drug interaction network **ogbl-ddi** (31). In Table 8, we summarize statistics of the above benchmark datasets. We compare against the following classic and state-of-the-art (SOA) baselines, including (i) MLP, (ii) GCN (35), (iii) Simplified Graph Convolution (SGC) (63), (iv) Graph Attention Networks (GAT) (59), (v) GraphSAGE (SAGE) (28), (vi) SEAL (68), (vii) Hyperbolic neural networks (HNN) (24), (viii) Hyperbolic Graph Convolutional Neural Networks (HGCN) (9), (ix) Persistence Enhanced Graph Neural Network (PEGN) (69), (x) Topological Loop-Counting Graph Neural Network (TLC-GNN) (64), (xi) BScNets (11), (xii) Neural Common Neighbor (NCN) (62), (xiii) PEG (60), (xiv) Cooperative Sheaf Neural Network (CSNN) (46), and (xv) Topology-Aware Graph Neural Network (TAGNN) (36).

For **trajectory prediction and fluid flow reconstruction tasks**, we consider the fluid flow data with the flow past a cylinder for 5 sensors (21) and compare SEN with 2 baselines, i.e., proper orthogonal decomposition (POD) and shallow decoder (SD) (21),

**Implementation Details.** We implement SEN with the Pytorch framework on one NVIDIA RTX A5000 GPUs with 24 GB RAM. Following (9), for all datasets of the task of link prediction, we randomly split edges into 85%/5%/10% for training, validation, and testing. For ogbl-ddi and ogbl-collab, we use the predefined edge ratios, i.e., 80%/10%/10% and 92%/4%/4% respectively. We also perform a one-sided two-sample $t$-test between the best result and the best performance achieved by the runner-up, where $*$ denotes statistically significant results. For protein datasets, we use 1120 chains in CATH datasets. TS50 comprises 50 protein chains, while TS500 consists of 500 randomly selected structures from the PISCES output. The learning rate of our SEN model is $1e^{-3}$, and the hidden layer dimension nhid is 128. We choose the dropout ratio as 0.1 and set the epoch as 100. For all datasets, SEN is trained by the Adam optimizer with the Cross Entropy Loss function. Code is available on `https://www.dropbox.com/scl/fo/6roa0i7wat4faeov46xfa/AH61vXbxJCNP8JVa71hUyRo?rlkey=po6htfm5yz53k56qakeiq9ebv&st=pxrjxu6j&dl=0`.

## 6.2 Results

On **forecasting** tasks, SEN delivers highly competitive gains for air quality prediction on both Eastern and Western U.S. (with significant and highly statistically significant margins, respectively) (Table 1 (*right*)). These findings are probably the most intuitive and could be expected as the role of air quality continuity on the boundary is the easiest to visualize.

Table 1: (*left*) Recovery rate of TS50, TS500, and CATH datasets and (*right*) prediction performance (relative $L_2$ error) for air quality ($PM_{2.5}$) in the Eastern and Western U.S. due to wildfires.

| Model | TS50 | TS500 | CATH | | Model | Eastern U.S. | Western U.S. |
|---|---|---|---|---|---|---|---|
| StructGNN (33) | 26.25% | 25.16% | 39.43% | | ResNet (30) | 75.91±2.63 | 220.70±0.96 |
| GCA (56) | 26.94% | 26.19% | 51.16% | | PDE-Net 2.0 (38) | 73.71±2.96 | 223.15±0.80 |
| PiFold (25) | 34.11% | 36.34% | 66.09% | | GNN-PDE (32) | 73.57±7.33 | 226.80±1.52 |
| GraDe-IF (66) | 14.64% | 14.59% | 11.71% | | SNN-PDE (13) | 64.42±2.36 | 200.66±0.91 |
| **SEN (ours)** | **34.93**% | **36.58**% | **66.40**% | | **SEN (ours)** | **62.72±2.44** | **197.27±0.03**$^*$ |

On **protein sequence recovery** tasks, Table 1 (*left*) shows that SEN consistently yields more competitive recovery rates in predicting protein structure, outperforming (albeit not substantially) even such a powerful benchmark as PiFold that is specifically designed for efficient protein inverse folding, and yielding up to 40% relative improvement with respect to the 3rd best model GCA.

On **link prediction** tasks, Table 2 indicates that: (i) Compared to the spectral-based ConvGNNs (i.e., GCN, SGC, and SIEG), SEN yields statistically significant improvements to the next best results for all 6 datasets; (ii) SEN outperforms spatial-based ConvGNNs (i.e., GAT, SAGE, SEAL, PEG, and NCN) with a significant margin; (iii) Compared to the hyperbolic-based NNs (i.e., HNN and HGCN), SEN improves upon HGCN by a margin of relative gains up to 14% and 14.47%, (iv) SEN further improves PH-based ConvGNNs (i.e., PEGN and TLC-GNN) with a significant margin on all 6 datasets; (v) Compared to the simplicial and topological neural networks (i.e., BScNets, SNN, SCNN, GSAN, MPNN, CSNN, and TAGNN), SEN delivers consistently competitive results. An interesting observation is comparison of SEN with sheaf neural networks, particularly, CSNN (46). We find that the highest relative gain of SEN vs. CSNN is achieved over the disease dataset (i.e., 11.51%). This phenomenon can be potentially explained by the latent hierarchical structure induced by the Susceptible-Infectious-Recovered (SIR) disease dynamics that the sheaf-based approach appear to be limited in capturing. In addition, Appendix D presents results on sensitivity of SEN and its competitors to the training size, suggesting that the delivered SEN performance is stable.

Furthermore, on **link prediction on large-scale networks**, Table 3 summarizes the performance results on ogbl-collab and ogbl-ddi datasets (31) with respect to one of the considered simplicial runner-ups (BScNets) and vanilla GCN. We find that SEN significantly outperforms baselines on both OGB data ($p$-value $< 0.05$). This indicates that our SEN with boundary condition operation generalizes well to large-scale graphs.

Finally, on **trajectory prediction and fluid flow reconstruction** tasks, as in the case of air quality, SEN does not only improve performance for these tasks but also tends to reduce variability (see Tables 5 and 4

Table 2: Link prediction (in ROC AUC).

| Model | Cora | PubMed | Meetings | Phone Calls | Airport | Disease |
|---|---|---|---|---|---|---|
| MLP | $83.15 \pm 0.51$ | $84.10 \pm 0.97$ | $63.20 \pm 6.22$ | $60.10 \pm 6.72$ | $89.51 \pm 0.52$ | $72.62 \pm 0.61$ |
| HNN (24) | $89.00 \pm 0.10$ | $94.87 \pm 0.11$ | $71.00 \pm 3.28$ | $60.90 \pm 4.25$ | $90.78 \pm 0.22$ | $75.10 \pm 0.35$ |
| GCN (35) | $90.42 \pm 0.28$ | $91.11 \pm 0.55$ | $72.08 \pm 4.19$ | $61.50 \pm 5.80$ | $89.27 \pm 0.42$ | $64.70 \pm 0.56$ |
| GAT (59) | $93.89 \pm 0.13$ | $91.22 \pm 0.12$ | $74.00 \pm 4.68$ | $63.40 \pm 5.20$ | $90.55 \pm 0.37$ | $69.99 \pm 0.32$ |
| SAGE (28) | $86.24 \pm 0.65$ | $85.96 \pm 1.16$ | $72.30 \pm 5.25$ | $62.07 \pm 5.49$ | $90.47 \pm 0.59$ | $65.91 \pm 0.33$ |
| SGC (63) | $91.67 \pm 0.20$ | $94.10 \pm 0.20$ | $73.38 \pm 3.49$ | $63.80 \pm 5.71$ | $90.01 \pm 0.32$ | $65.21 \pm 0.23$ |
| SEAL (68) | $92.55 \pm 0.50$ | $92.42 \pm 0.12$ | $71.09 \pm 7.50$ | $62.96 \pm 4.17$ | $95.16 \pm 0.39$ | $85.23 \pm 0.79$ |
| HGCN (9) | $93.00 \pm 0.45$ | $96.29 \pm 0.18$ | $83.20 \pm 4.15$ | $70.20 \pm 3.77$ | $96.40 \pm 0.19$ | $90.80 \pm 0.30$ |
| PEGN (69) | $93.13 \pm 0.50$ | $95.82 \pm 0.20$ | $74.17 \pm 5.00$ | $65.23 \pm 4.15$ | $95.46 \pm 0.71$ | $83.61 \pm 1.26$ |
| TLC-GNN (64) | $94.22 \pm 0.78$ | $97.03 \pm 0.10$ | $73.20 \pm 5.32$ | $66.17 \pm 3.90$ | $96.60 \pm 0.69$ | $86.19 \pm 1.23$ |
| SNN (17) | $53.55 \pm 1.24$ | $65.89 \pm 2.33$ | $54.93 \pm 6.13$ | $44.54 \pm 1.48$ | $66.25 \pm 1.70$ | $51.31 \pm 5.48$ |
| SCNN (65) | $71.73 \pm 4.42$ | $78.74 \pm 1.46$ | $70.06 \pm 5.59$ | $71.46 \pm 1.79$ | $90.54 \pm 1.44$ | $54.62 \pm 4.41$ |
| BScNets (11) | $94.90 \pm 0.70$ | $97.55 \pm 0.12$ | $88.05 \pm 5.51$ | $79.43 \pm 6.04$ | $97.57 \pm 0.67$ | $98.60 \pm 0.58$ |
| SIEG (51) | $93.92 \pm 0.05$ | $92.10 \pm 0.44$ | $82.85 \pm 3.21$ | $78.94 \pm 0.41$ | $95.24 \pm 0.38$ | $91.35 \pm 0.90$ |
| PEG (60) | $94.46 \pm 0.34$ | $96.98 \pm 0.35$ | $88.21 \pm 3.05$ | $77.87 \pm 0.28$ | $93.10 \pm 0.87$ | $98.42 \pm 0.22$ |
| NCN (62) | $94.72 \pm 0.70$ | $96.25 \pm 0.19$ | $88.55 \pm 2.66$ | $76.74 \pm 0.31$ | $94.21 \pm 0.27$ | $98.75 \pm 0.56$ |
| GSAN (3) | $91.71 \pm 0.49$ | $95.30 \pm 0.61$ | $87.31 \pm 7.31$ | $79.20 \pm 0.33$ | $95.97 \pm 0.42$ | $90.02 \pm 0.21$ |
| MPNN (18) | $92.15 \pm 0.20$ | $92.27 \pm 0.50$ | $85.94 \pm 2.58$ | $77.15 \pm 0.89$ | $95.40 \pm 0.72$ | $96.95 \pm 0.79$ |
| CSNN (46) | $93.68 \pm 0.42$ | $95.45 \pm 0.22$ | $88.63 \pm 4.27$ | $78.52 \pm 2.94$ | $96.11 \pm 0.52$ | $88.74 \pm 0.57$ |
| TAGNN (36) | $92.33 \pm 0.41$ | $95.35 \pm 0.15$ | $87.92 \pm 3.88$ | $76.87 \pm 2.61$ | $96.12 \pm 0.31$ | $92.56 \pm 0.44$ |
| **SEN (ours)** | $\mathbf{95.00 \pm 0.39}$ | $\mathbf{97.73 \pm 0.32}$ | $\mathbf{89.12 \pm 3.82}$ | $\mathbf{80.36 \pm 0.34}$ | $\mathbf{97.67 \pm 0.51}$ | $\mathbf{98.95 \pm 0.16}$ |

Table 3: Link prediction on large-scale graphs (in ROC AUC).

| Method | ogbl-ddi | ogbl-collab |
|---|---|---|
| SEN | $\mathbf{32.31 \pm 0.09}^*$ | $\mathbf{31.62 \pm 0.07}^*$ |
| BScNets | $31.53 \pm 0.08$ | $31.00 \pm 0.08$ |
| GCN | $25.13 \pm 0.06$ | $27.08 \pm 0.05$ |

respectively). Again, these phenomena could be expected, given the intrinsic role of the continuity constraints in the underlying physics of fluid flow dynamics.

Table 4: Fluid flow data reconstruction (in NME).

| Dataset | SEN | POD | POD PLUS | SD |
|---|---|---|---|---|
| Fluid flow | **0.004** | 0.488 | 0.016 | 0.006 |

**Ablation Study and Sensitivity Analysis.** Table 6 presents the ablation study on the individual SEN components. We find that SEN outperforms all ablated variants SEN w/o *continuity* conditions (i.e., SEN w/o CC), SEN w/o attention mechanism (i.e., SEN w/o Attn), and SEN w/o combinatorial Laplacian (SEN w/o CL). While the continuity conditions play important roles in all considered scenarios, their contribution is highest for prediction of air quality, and this phenomenon could be intuitively expected given the physics underlying transformation of pollutants in the atmosphere. Differences between SEN and its three variants are all highly statistically significant ($p$-value $< 0.05$) on Western U.S. data. On the Synthetic Flow dataset, we also obtain a highly statistically significant result between SEN and SEN w/o CL ($p$-value $< 0.01$), a statistically significant result between SEN and SEN w/o CC ($p$-value $< 0.05$), and a significant result between SEN and SEN w/o Attn ($p$-value $< 0.1$). Table 7 evaluates the sensitivity of SEN to the fusion coefficients $\rho_G$ and $\rho_S$, which control the relative contributions of graph-based and simplicial representations. Overall, SEN

Table 5: Trajectory classification accuracy (in %).

| Dataset | MPSN with Tanh | GSAN | MPNN | SEN (ours) |
|---|---|---|---|---|
| Synthetic Flow | 95.20±1.80 | 89.55±2.49 | 95.28±2.31 | **96.70±1.50** |

demonstrates strong robustness across a broad range of settings. The results indicate that SEN consistently achieves competitive performance regardless of the specific fusion weights, suggesting that the learned graph and simplicial representations provide complementary information, e.g., the best performance is generally obtained when the two components are balanced (i.e., $\rho_G = \rho_S = 0.5$ for Cora, Meetings, and Disease) and the performance degradation under neighboring configurations is minimal.

Finally, we would like to highlight what higher order simplices bring (albeit formally this falls outside the ablation study and refers more generally to all simplicial neural networks rather than SEN in particular). By reducing $p$ to 0, we arrive to the node level and can compare SEN and other simplicial models to vanilla GCN. While SEN and other simplicial approaches substantially outperform GCN for all datasets, the highest relative gains of 30% and 50% for SEN vs. GCN are observed for the Phone Calls of Sicilian Mafia and Disease data, respectively, followed by 20% relative gains for the Meetings of Sicilian Mafia. This is in fact not surprising as such data are characterized by the intrinsic hierarchical structure and higher order interconnectivity due to the latent social interaction patterns (6; 7; 10).

Table 6: Ablation study for link prediction (in ROC AUC).

| | Cora | Meetings | Phone Calls | East. U.S. | West. U.S. | Synthetic Flow |
|---|---|---|---|---|---|---|
| **SEN** | **95.00±0.39*** | **89.12±3.82** | **80.36±0.34** | **62.72±2.44** | **197.27±0.03*** | **96.70±1.50*** |
| **w/o CC** | 93.72±0.69 | 88.77±5.12 | 79.35±0.22 | 64.10±1.22 | 207.18±0.36 | 95.08±1.70 |
| **w/o Attn** | 93.10±0.35 | 87.28±2.74 | 79.99±0.81 | 63.79±2.31 | 201.63±0.10 | 95.68±1.25 |
| **w/o CL** | 93.85±0.74 | 85.96±2.03 | 76.80±0.77 | 62.94±2.57 | 199.75±0.15 | 93.39±2.38 |

Table 7: Sensitivity analysis on SEN (in ROC AUC).

| $\rho_\mathcal{G}/\rho_\mathcal{S}$ | Cora | Meetings | Phone Calls | Disease |
|---|---|---|---|---|
| 0.3 / 0.7 | $94.61 \pm 0.45$ | $88.55 \pm 4.00$ | $79.72 \pm 0.57$ | $98.22 \pm 0.31$ |
| 0.4 / 0.6 | $94.88 \pm 0.41$ | $88.91 \pm 3.95$ | $80.08 \pm 0.36$ | $98.73 \pm 0.24$ |
| 0.5 / 0.5 | $\mathbf{95.00 \pm 0.39}$ | $\mathbf{89.12 \pm 3.82}$ | $80.11 \pm 0.37$ | $\mathbf{98.95 \pm 0.16}$ |
| 0.6 / 0.4 | $94.93 \pm 0.43$ | $88.97 \pm 3.87$ | $\mathbf{80.36 \pm 0.34}$ | $98.82 \pm 0.21$ |
| 0.7 / 0.3 | $94.54 \pm 0.48$ | $88.43 \pm 4.15$ | $79.67 \pm 0.42$ | $98.36 \pm 0.35$ |

**Computational Complexity.** Computational complexity of SEN is $O(\mathcal{N} + 2\mathcal{M} + \mathcal{Q} + L(\mathcal{M}d + \mathcal{N}d^2))$, where $\mathcal{N}$ is the number of 0-simplices, $\mathcal{M}$ is the number of 1-simplices, $\mathcal{Q}$ is the number of 2-simplices $d$ is the number of hidden dimensions, and $L$ is the number of layers. For higher-order simplices, boundary matrices $\mathbf{B}_1$ and $\mathbf{B}_2$ can be calculated efficiently with the computational complexity $\mathcal{O}(\mathcal{N} + \mathcal{M})$ and $\mathcal{O}(\mathcal{M} + \mathcal{Q})$, respectively. We further compare SEN with strong baselines from the perspective of model size. On Cora, using the same hidden dimension (i.e., $d = 128$), SEN contains around 0.23M trainable parameters, which is comparable to GCN (0.18M) and GAT (0.27M). The additional continuity-aware module in SEN introduces only a lightweight set of trainable transformations, since the main higher-order propagation is performed through sparse boundary operators and combinatorial Laplacians. In practice, SEN is not restricted to explicit enumeration of all high-dimensional simplices. Similar to other simplicial and topological learning methods, one can employ various sparsification approaches, witness and Dowker complexes (15; 16), local simplex construction (e.g., $k$-hop neighborhood-based simplex construction), or sampling-based strategies to further reduce computational overhead. Furthermore, while traditional simplicial neural networks typically rely on enumeration of simplices, this step can be relaxed by using, for instance, lazy witness complexes or by utilizing cell complexes (55). Moreover, the proposed module naturally generalizes to higher dimensions by

replacing the 2-simplex interior boundary operator with its $p$-dimensional counterpart, allowing continuity constraints to be enforced on shared $(p-1)$-faces.

## 7 Conclusion

We have proposed a new Simplicial Element Network (SEN) framework with the dual simplicial convolution encoder to enhance representation learning of higher order graph (sub)structures. In particular, motivated by the numerical methods for PDEs, we have introduced the ideas of finite element methods for PDE solvers to simplicial convolution on graphs and have explicitly enforced the natural discrete-valued analogue of the *continuity* condition for a case of neighboring simplices. The SEN approach is both tractable and versatile, yielding promising results in a wide range of downstream tasks, from protein sequence recovery and spatio-temporal forecasting to link prediction and trajectory tracking. Furthermore, while in the current paper for simplicity we have focused on the 2-simplices, the SEN methodology is extendable for higher order simplices.

Several promising research directions remain open. In particular, SEN could be expanded to incorporate a broader class of boundary conditions, such as derivative consistency and other FEM-inspired constraints on simplicial interfaces. Such extensions may be approached through construction of the appropriate trainable local shape functions or via the weak enforcement mechanisms based on penalty formulations, similarly to those used in discontinuous Galerkin methods (42) and variational inference (26). Another important avenue for future research is the theoretical analysis of SEN, particularly, with regard to its expressive power.

More broadly, we believe that a stronger integration of graph representation learning and numerical PDE methods offers multiple new opportunities for both fields, from data- and task-tailored neural network architectures to graph diffusion to more efficient physics-informed machine learning and scientific computing. SEN represents just one of the first steps toward establishing such connection, highlighting the potential of FEM-inspired principles for advancing topological and geometric deep learning.

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

# A    Additional Details on Simplicial Complexes, Homology and Laplacians

For a $p$-simplex $\sigma$ of $p > 0$, we can also define its *orientation* by selecting some order for its elements, and two orderings are equivalent if they differ by an even permutation.

**Definition A.1.** For an oriented simplicial complex $\mathcal{K}$, its two $p$-dimensional simplices $\sigma^1$ and $\sigma^2$ are similarly oriented and denoted as $\sigma^1 \sim \sigma^2$, if they are lower adjacent and consistently oriented on on the common lower $(p-1)$-simplex. Two simplices $\sigma^1$ and $\sigma^2$ are dissimilarly oriented and denoted as $\sigma^1 \nsim \sigma^2$, if they are lower adjacent but have different signs on the common lower $(p-1)$-simplex.

**Definition A.2.** Formally, a simplicial complex $\mathcal{K}$ is a finite set of simplices that is closed under taking subsets, i.e. $\mathcal{K}$ is to satisfy two essential conditions: 1) Any face of a simplex from $\mathcal{K}$ is also in $\mathcal{K}$; 2) Intersection of any two simplices in $\mathcal{K}$ is either empty or shares faces. If a $p-1$-dimensional simplex $\mu$ is a face of $p$-dimensional simplex $\sigma$, we denote this relation as $\mu < \sigma$. Here $p \in \mathbb{Z}^+$ and $0 \le p \le dim(\mathcal{K})$.

For a simplicial complex $\mathcal{K}$, its $p$-dimensional chain group $C_p(\mathcal{K})$ is composed of $p$-simplices in $\mathcal{K}$. Let $[v_0, v_1, \cdots v_p]$ be a $p$-simplex $\sigma^p$, the weighted boundary operator $\partial_p : C_p \to C_{p-1}$ for a given $\sigma^p \in \mathcal{K}$ is defined as

$$\partial_p(\sigma^p) = \sum_{i=0}^{p} (-1)^i [v_0, v_1, \cdots, \hat{v}_i, \cdots, v_p].$$

Here the boundary of $p$-simplex is made of $(p-1)$-simplices $[v_0, v_1, \cdots, \hat{v}_i, \cdots, v_p]$, where $\hat{v}_i$ means that $v_i$ has been removed from the sequence $v_0, \cdots, v_p$. It is also well-known that $\partial_{p-1}\partial_p = 0$. That is, the unweighted boundary operator $\partial_p : C_p \to C_{p-1}$ for a given $\sigma^p \in \mathcal{K}$ is defined as

$$\partial_p(\sigma^p) = \sum_{i=0}^{p} (-1)^i [v_0, v_1, \cdots, \hat{v}_i, \cdots, v_p].$$

For an oriented simplicial complex $\mathcal{K}$, its two oriented $p$-dimensional simplices $\sigma_1$ and $\sigma_2$ are similarly oriented and denoted as $\sigma_1 \sim \sigma_2$, if they are lower adjacent and have the same sign on the common lower $(p-1)$-simplex. Two simplex $\sigma_1$ and $\sigma_2$ are dissimilarly oriented and denoted as $\sigma_1 \nsim \sigma_2$, if they are lower adjacent but have different signs on the common lower $(p-1)$-simplex.

The $p$-th cycle group $Z_p$ is defined as

$$Z_p = \ker(\partial_p) = \{c \in C_p | \partial_p(c) = 0\},$$

and $p$-th boundary group $B_p$ is

$$B_p = \operatorname{im}(\partial_{p+1}) = \{c \in C_p | \exists d \in C_{p+1} : c = \partial_{p+1}(d)\}.$$

In turn, the $p$-th homology group is defined as $H_p = Z_p/B_p$. Its rank is $p$-th Betti number that satisfies

$$\beta_p = \operatorname{rank} H_p = \operatorname{rank} Z_p - \operatorname{rank} B_p.$$

With the boundary operators, we have chain complexes

$$\cdots \xrightarrow{\partial_{p+2}} C_{p+1} \xrightarrow{\partial_{p+1}} C_p \xrightarrow{\partial_p} C_{p-1} \xrightarrow{\partial_{p-1}} \cdots$$

The adjoint of $\partial_p$, which is

$$\partial_p^* : C_{p-1} \to C_p,$$

satisfies the inner product relation $\langle \partial_p(f), g \rangle = \langle f, \partial_p^*(g) \rangle$, for every $f \in C_p$, $g \in C_{p-1}$.

Furthermore, if we consider an unoriented counterpart $\tilde{\mathcal{K}}$ of simplicial complex $\mathcal{K}$, then the incidence matrix $\mathcal{B}_p$ reduced to $\tilde{\mathcal{B}}_p$ such that $\tilde{\mathcal{B}}_p(i,j) = 1$ if $\sigma_i^{p-1} < \sigma_j^p$ and 0, otherwise.

**Lower Laplacian on simplicial complexes** Note that entries of $\mathbf{L}_p^{\text{down}}$ are given by

$$
\mathbf{L}_p^{\text{down}}(i,j) = \begin{cases} deg_L(\sigma_i^p), & i = j \\ 1, & i \neq j, \sigma_i^p \smile \sigma_j^p, \sigma_i^p \sim \sigma_j^p \\ -1, & i \neq j, \sigma_i^p \smile \sigma_j^p, \sigma_i^p \nsim \sigma_j^p \\ 0, & i \neq j \text{ and } \sigma_i^p \not\smile \sigma_j^p. \end{cases}
$$

Here $deg_L(\sigma_i^p)$ is the lower degree of $\sigma_i^p$ in $\mathcal{K}$, i.e., the number of nonempty $(p-1)$-simplices in $\mathcal{K}$ that are faces of $\sigma_i^p$. Hence, $\mathbf{L}_p^{\text{down}}$ allows us to explicitly record the boundaries shared by two $p$-dimensional simplices $\sigma_i^p$ and $\sigma_j^p$, i.e., whenever $\sigma_i^p$ and $\sigma_j^p$ share a face and are similarly oriented, the entry $\mathbf{L}_p^{\text{down}}(i,j) = 1$; if $\sigma_i^p$ and $\sigma_j^p$ share a face but are dissimilarly oriented, $\mathbf{L}_p^{\text{down}}(i,j) = -1$.

**Upper Laplacian on simplicial complexes** Entries of $\mathbf{L}_p^{\text{up}}$ are given by

$$
\mathbf{L}_p^{\text{up}}(i,j) = \begin{cases} deg_U(\sigma_i^p), & i = j \\ -1, & i \neq j, \sigma_i^p \frown \sigma_j^p, \sigma_i^p \sim \sigma_j^p \\ 1, & i \neq j, \sigma_i^p \frown \sigma_j^p, \sigma_i^p \nsim \sigma_j^p \\ 0, & i \neq j \text{ and } \sigma_i^p \not\frown \sigma_j^p. \end{cases}
$$

Here $deg_U(\sigma_i^p)$ is the upper degree of $\sigma_i^p$ in $\mathcal{K}$, i.e., the number of nonempty $(p+1)$-simplices in $\mathcal{K}$ that are faces of $\sigma_i^p$.

**Combinatorial Laplacian or discrete Hodge-Laplacian on graphs** Entries of $\mathbf{L}_p$ are given by

$$
\mathbf{L}_p(i,j) = \begin{cases} d(\sigma_i^p), & i = j \\ 1, & i \neq j, \sigma_i^p \not\frown \sigma_j^p, \sigma_i^p \smile \sigma_j^p, \sigma_i^p \sim \sigma_j^p \\ -1, & i \neq j, \sigma_i^p \not\frown \sigma_j^p, \sigma_i^p \smile \sigma_j^p, \sigma_i^p \nsim \sigma_j^p \\ 0, & i \neq j \text{ and either } \sigma_i^p \frown \sigma_j^p \text{ or } \sigma_i^p \not\smile \sigma_j^p. \end{cases}
$$

Here $d(\sigma_i^p) = deg_U(\sigma_i^p) + p + 1$ (where $deg_U(\sigma_i^p)$ denotes the upper degree of $\sigma_i^p$). In addition to $\mathbf{L}_p$ which is typically used in SNNs, we can use both lower and upper combinatorial Laplacians $\mathbf{L}_p^{\text{down}}$ and $\mathbf{L}_p^{\text{up}}$ to define the associated boundary conditions and simplex message passing mechanisms.

# B Proofs

**Proof of Proposition 4.3.** From *Definition 0.5*, for all $z \in \mathbb{R}^n$, $||z|| \neq 0$, $z^\top \mathbf{L}_p^{\text{down}} z = (\mathbf{B}_p z)^\top (\mathbf{B}_p z) \geq 0$ and $(\mathbf{L}_p^{\text{down}})^\top = (\mathbf{B}_p^\top \mathbf{B}_p)^\top = \mathbf{B}_p^\top \mathbf{B}_p = \mathbf{L}_p^{\text{down}}$. Derivations for $\mathbf{L}_p^{\text{up}}$ are analogous.

**Proof of Proposition 4.5.** Let $\tilde{\mathcal{B}}_p = \{\tilde{b}, \dots, \tilde{b}_n\}$, where $\tilde{b}_i$ is an $m$-dimensional vector in $\mathbb{R}^m$. Hence, $\tilde{\mathbf{L}}_p^{\text{down}}(k,j) = <\tilde{b}_k, \tilde{b}_j>$, where $<\cdot,\cdot>$ is an inner product. If $k \neq j$, $\sigma_i^{p-1} < \sigma_j^p$ and $\sigma_i^{p-1} < \sigma_k^p$, that is, $\sigma_j^p$ and $\sigma_k^p$ are lower adjacent, then the $i$-th component of $\tilde{b}_k$ and $\tilde{b}_j$ is 1. If $k \neq j$ and $\sigma_i^{p-1} \not< \sigma_j^p$ (or $\sigma_i^{p-1} \not< \sigma_k^p$), then the $i$-th component of $\tilde{b}_k$ ($\tilde{b}_j$, respectively) is 0. If $k = j$, then

$$
\tilde{\mathbf{L}}_p^{\text{down}}(j,j) = <\tilde{b}_j, \tilde{b}_j> = \sum_i \mathbb{1}_{\sigma_i^{p-1} < \sigma_j^p} = \deg_{\text{L}}(\sigma_j^{\text{p}})
$$

$$
= \sum_{k=1, k \neq j}^{n} |\tilde{\mathbf{L}}_p^{\text{down}}(k,j)|, \tag{3}
$$

which implies that $\tilde{\mathbf{L}}_p^{\text{down}}$ is diagonally dominant.

Positive semi-definitess of $\tilde{\mathbf{L}}_p^{\text{down}}$ follows verbatim *Proposition 0.7*.

## C  Additional Details on Experimental Settings

We report the average and standard deviation of ROC AUC values. For all methods, we run 5 times with the same partition and report the average and standard deviation of accuracies. The learning rate is searched in $\{1e^{-5}, 1e^{-4}, 1e^{-3}, 1e^{-2}, 1e^{-1}\}$, and the hidden layer dimension nhid is searched in $\{4, 8, 16, 32, 64, 128\}$. The number of simplicial convolution operations is set as $\{1, 2, 3, 4, 5\}$ and we choose the dropout ratio as 0.5 for all datasets. The tuning of SEN on each dataset is done via the grid hyperparameter configuration search over a fixed set of choices and the same cross-validation setup is used to tune the baselines. For a fair comparison, the results of MLP, HNN, GCN, GAT, SAGE, SGC, SEAL, HGCN, PEGN, TLC-GNN, SNN, and BScNets are directly reported from their original papers. For the remaining methods, we obtained the results using the authors' publicly available open-source implementations.

Table 8: Summary of datasets used in link prediction tasks.

| Dataset | # Nodes | # Edges | # Features |
|---|---|---|---|
| Cora | 27,08 | 5429 | 1,433 |
| PubMed | 19,717 | 88,651 | 500 |
| Meetings | 101 | 256 | 4 |
| Phone Calls | 100 | 124 | 4 |
| Airport | 3,188 | 18,631 | 4 |
| Disease | 1,044 | 1,043 | 1,000 |
| ogbl-ddi | 4,267 | 1,334,889 | - |
| ogbl-collab | 235,868 | 1,285,465 | 128 |

## D  Additional Experiments on Sensitivity to the Training Size

We have also conducted experiments on SEN, NCN, and BScNets on Cora and Disease datasets with various training sizes, i.e., 60%, 70%, and 80%.

Table 9 demonstrates that (i) on Cora, SEN outperforms both BScNets and NCN on all three different training sizes, with significance, statistical significance or high statistical significance. Specifically, on 60% training size, SEN achieves statistically significant results with respect to BScNets and to NCN; on 70% training size, SEN achieves significant results with respect to BScNets and highly statistically significant results with respect to NCN; on 80% training size, SEN achieves significant results with respect to BScNets and highly statistically significant results with respect to NCN. (ii) on Disease, compared to BScNets and NCN, SEN always achieves either statistically or highly statistically significant results on all three training sizes; Specifically, on 60% training size, SEN yields highly statistically significant results with respect to BScNets and to NCN; on 70% training size, SEN yields highly statistically significant results with respect to BScNets and to NCN; on 80% training size, SEN yields highly statistically significant results with respect to BScNets and statistically significant results with respect to NCN. These findings, along with the formal statistical inference on the delivered performance results, indicate that SEN outperforms BScNets and NCN not by chance, and the SEN performance gains are consistent across varying sample sizes.

Table 9: Performance comparison under different proportions of labeled data.

| Model | 60% | | 70% | | 80% | |
| --- | --- | --- | --- | --- | --- | --- |
| | **Cora** | **Disease** | **Cora** | **Disease** | **Cora** | **Disease** |
| BScNets | $91.32 \pm 0.95$ | $96.32 \pm 0.59$ | $92.90 \pm 0.73$ | $96.61 \pm 0.37$ | $93.19 \pm 0.76$ | $97.12 \pm 0.68$ |
| NCN | $91.55 \pm 0.62$ | $96.27 \pm 1.02$ | $92.30 \pm 0.35$ | $96.92 \pm 0.53$ | $93.17 \pm 0.43$ | $97.70 \pm 0.89$ |
| **SEN (ours)** | $\mathbf{92.45 \pm 0.68^*}$ | $\mathbf{98.20 \pm 0.65^*}$ | $\mathbf{93.62 \pm 0.71^*}$ | $\mathbf{98.25 \pm 0.25^*}$ | $\mathbf{94.21 \pm 0.65^*}$ | $\mathbf{98.62 \pm 0.49^*}$ |

