# OpenReview forum: "Keep Your Boundaries: From Finite Elements to Simplicial Convolution"
_TMLR — Under review for TMLR_

### Review · Reviewer_EKyR · 2026-07-06

**Summary Of Contributions:**

This paper proposes a graph learning architecture inspired by finite element methods. The main idea is to extend simplicial neural networks by introducing a continuity-aware message passing mechanism over interior boundaries, implemented through an interior boundary matrix and the corresponding interior upper combinatorial Laplacian. The model combines standard simplicial convolution with this additional continuity-oriented branch, and evaluates the resulting architecture on protein sequence recovery, link prediction, air quality forecasting, trajectory prediction, and fluid flow reconstruction.
The paper addresses an interesting problem. The connection between finite element ideas, simplicial complexes, Hodge Laplacians, and graph learning is meaningful, and the empirical evaluation is broad. The results are generally positive, and the ablation study suggests that the proposed continuity-aware component contributes to performance. That being said, there are some issues (discussed below).

**Additional Comments:**

None.

**Audience:**

Yes

**Audience Explanation:**

The paper is relevant to several parts of the TMLR audience: graph neural networks, topological deep learning, geometric deep learning, PDE-inspired architectures, and scientific machine learning. The idea of importing FEM-style inductive biases into simplicial graph learning is interesting and worth studying.

**Broader Impact Concerns:**

I do not see specific broader impact concerns that require additional discussion.

**Claims And Evidence:**

No

**Claims Explanation:**

The central claim is overstated. The paper repeatedly frames the method as enforcing FEM-style continuity constraints, but the actual implementation is a special message passing operation over shared interior simplices. The authors explicitly state that they do not define the boundary operator and instead use this message passing module as an approximation of continuity constraints. This is a useful inductive bias, but it is not the same as enforcing continuity. The paper also does not directly measure whether boundary mismatch is reduced. Therefore, the main conceptual claim is not fully supported by the presented theory or experiments.

The issue is with the main methodological claim. The paper claims to enforce continuity constraints in the FEM sense. However, the method does not impose an equality constraint on boundary values, does not define an explicit boundary value operator, and does not optimize a boundary mismatch penalty. Instead, it adds a message passing module over edges or faces shared by multiple higher-order simplices. This can be reasonably described as continuity-aware or continuity-inspired message passing, but not as explicit enforcement of continuity.
This distinction matters because the FEM analogy is the main conceptual contribution of the paper. The current evidence shows that the added module improves predictive performance in several tasks, but it does not show that the learned representation satisfies continuity, or even that the boundary mismatch is reduced relative to the ablated model.


The theory is also not strong enough to support the main claim. The propositions mostly establish standard properties of combinatorial Laplacians and message passing operators. They do not prove continuity enforcement, stability, expressivity, approximation properties, or a formal relation to FEM discretization quality.
The experimental section is broad, but the fairness of the comparisons is not fully clear. Some baselines appear to be taken from prior papers, while others are run by the authors. The paper should clearly state whether all baselines use the same splits, preprocessing, features, evaluation protocol, and tuning budget. Without that, the strength of the empirical comparison is harder to assess.

**Requested Changes:**

- The paper must correct or substantiate the claim that SEN “enforces” continuity constraints. As written, the method appears to approximate continuity through a special message passing operation on shared interior simplices. This is not equivalent to enforcing FEM-style continuity. The authors should either weaken the language throughout the paper to “continuity-aware” or “continuity-inspired”, or provide a precise mathematical formulation showing how continuity is enforced.

- The paper should include a direct boundary-consistency experiment. The most natural test is to measure boundary mismatch between adjacent simplices sharing a face, before and after applying the proposed continuity module, and compare SEN against SEN without the continuity component and against existing simplicial neural networks. This experiment is important because it directly tests the central claim of the paper.

- The method section should clarify how the continuity branch is used in the final representation. The equations introduce an additional continuity output, but the final attention/fusion equation appears to omit this output and only combines the standard simplex embeddings and an edge-level embedding. This makes the actual architecture ambiguous.

- The authors should clarify the experimental protocol for all baselines. The paper should explicitly state which baselines were rerun by the authors and which numbers were copied from prior papers. It should also state whether the same splits, features, preprocessing, evaluation metrics, and tuning budget were used. This is important because the paper compares against many baselines across heterogeneous tasks.

- The statement that all datasets are trained with cross-entropy loss should be corrected or clarified. This is inconsistent with regression/reconstruction tasks such as air quality forecasting and fluid flow reconstruction.

- The theoretical section should be made more substantive. The current propositions are mostly standard properties of Laplacian operators. A stronger paper would include a result connecting the proposed module to reduced boundary mismatch, stability, expressivity, or a clearer FEM-style variational interpretation.

- The computational cost discussion should include the cost and memory overhead of constructing simplices and boundary matrices, not only the complexity of the message passing layers and parameter count.

---

### Review · Reviewer_kxh3 · 2026-07-10

**Summary Of Contributions:**

### **Strength**
- Well-motivated problems setting. Introduction is also well-organized and mostly convincing.
- A wide range of experimental scenarios. GNN models involved are almost exhaustive.


### **Weakness**

**Inaccurate supports and/or evidences of their standpoint**
- "Neither of these simplicial methods ...": The latter, sheaf NNs, is technically not simplicial methods.
- "Here f is the node feature": Distinction between original or latent does matter.

**Weak theoretical rigor for the paper's contribution**
-  I do not find anywhere in this paper proving that the proposed method successfully impose the continuity constraint. Furthermore, even if I favourbly consider all the proposed Laplacians as a main contribution to impose the constraint, this sounds contradictory to me because this paper https://arxiv.org/pdf/2010.03633 already incorporates graph Hodge laplacians to define message passing. This makes it uncertain that the performance gain is coming from the claimed contribution.

**Questionable experimental settings**
- The performance of proposed framework in test dataset highly depends on the orientation given for simplicies. Even when the proposed model is trained on training datasets, the performance is orientation dependent and its performance can vary significantly depending on the orientation given to simplicies in test dataset. The model's robustness to the orientation should be evaluated more thoroughly.
-  The ablation study for $\tilde{L}$ looks missing. Therefore, I am not yet convinced with the significance of this claim "However, this scheme may not necessarily highlight and learn certain p-dimensional simplices which receive messages from multiple (p + 1)-dimensional simplices, e.g., cofaces. For instance, when p = 1 (i.e., 1-dimensional simplices/edges), edges are shared by more than one triangle."

**Loose and misleading description of theoretical contributions**
- Orientation and its signs have to be explained systematically. Otherwise extremely hard to reproduce all the experiments.
- The statement of Proposition 4.4 is not very clear. Judging from what follows, this is rather a definition, not a theoretically deduced result.
- $\mathcal{B}$ in Proposition 4.5 is not defined properly, but the proof suddenly assumes $b_{i}$.

**Mathematically loose notations and terminologies**
- definition of "$σ_1$ ∼ $σ_2$," in Def A.1 sounds in prose. "Consistent orientation" essentially does not explain anything.
- $σ^{p−1}_{i}$ ∼ $σ^p$ in Def 4.1. is not defined, while relation ∼ is defined when these two simplicies have the same dimension.
- The domain and codomain of each map are very unclear. Better to clarify the dimension and size of these domains.
- What the upperscripts of $\sigma^{1}$ and $\sigma^{2}$ mean is different from that of $\sigma^{p}$

**Minor**: I found a couple of typos.
- Clearly, air quality is on the border of Quebec and New York state ought to be the same
- What is propostion 0.7????
- Definition 4.2:L^{up} is duplicated

**Question**:
- How is $\tilde{Z}^{(\ell)}_{1, \mathcal{S}}$ in the equation (2) defined? $\hat{H}^{(\ell + 1)}$ looks ill-defined because $H^{(\ell + 1)}$ is already zero dimensional.
- The number of (P + 1) embeddings looks P+2, not P+1
- What is \mathcal{S}? Is it just a symbol or mathematically meaningful object?

**Additional Comments:**

Not particularly.

**Audience:**

No

**Audience Explanation:**

As I mentioned in the previous column, it is unclear whether imposing the continuous constraint actually helps boost the performance. The motivation is written very well so TMLR's audience could get interested, but the paper will highly likely leave them uncertain that the cliamed issue is a main bottleneck in improving the model's performance.

**Broader Impact Concerns:**

No concerns on the ethical implications.

**Claims And Evidence:**

No

**Claims Explanation:**

It is not clear that the continuity constraint is successfully imposed from the current description of the paper. The difference between the prior relevant method as mentioned above and the proposed method is also not clarified. Therefore, it is not very clear which part in the proposed method contributes to the reporeted perofrmance gain.

**Requested Changes:**

Please consider weak points in the weakness section above as requested changes or points to address.

---

### Review · Reviewer_RpEC · 2026-07-13

**Summary Of Contributions:**

This paper proposes the Simplicial Element Network, bridging finite element continuity and simplicial graph learning via an interior boundary matrix and a continuity-aware encoder. Despite its engaging cross-disciplinary motivation and extensive evaluation across five distinct domains, the paper falls short in technical execution. The central continuity concept is not mathematically implemented as a constraint, the encoder houses unresolved dimensional and indexing errors, and the experimental design fails to convincingly validate the reported performance gains against comparable baselines.

**Audience:**

Yes

**Audience Explanation:**

In principle, connecting finite element interface conditions with higher-order message passing is a highly valuable pursuit for the geometric DL and SciML communities. However, because the current manuscript fails to provide a mathematically sound formulation or a trustworthy empirical evaluation, it cannot be accepted in its present form.

**Claims And Evidence:**

No

**Claims Explanation:**

The paper falls short of the required standard due to a major disconnect between its primary claim and actual implementation. While the authors assert that SEN enforces finite-element-style continuity across shared simplex boundaries, Section 5 explicitly states that the model lacks both a boundary operator and a formula for boundary values, relying instead on message passing over selected shared edges as a proxy. Although this operation alters the representations of interior edges, it neither defines a boundary trace nor proves that two adjacent simplex functions match on their common face. Consequently, the framework fails to establish the foundational property driving its novelty and physical grounding. This theoretical deficiency is compounded by critical empirical flaws: incomplete task definitions, potential data leakage in link prediction via simplex construction, baseline results improperly imported from external studies with differing protocols, flawed complexity accounting, and statistical tests that are inadequate for the scope of the reported comparisons.

**Requested Changes:**

1. The authors should substitute the current informal narrative with a rigorous mathematical framework that explicitly defines the boundary trace operator and the exact continuity or variational constraints imposed across shared element interfaces.

2. The entire encoder architecture, including Definition 4.6 and Equation 2, should be rederived to establish strict dimensional and type consistency, ensuring that the row and column spaces of all boundary and incidence matrices are explicitly stated, the general dimensional forms match the edge triangle instances, and the computation graph contains no dimensionally undefined tensor multiplications.

3. The theoretical foundations should be thoroughly repaired by either aligning Proposition 4.5 and Appendix B with the actual operator utilized in the methodology, resolving the reliance on non-existent references, and correcting the discrepancy between the lower and upper Laplacians, or removing the unsubstantiated claims regarding finite element compatibility entirely.

4. The experimental protocol for the link prediction task must explicitly demonstrate that all higher-order topological structures, including simplicial complexes and incidence matrices, are constructed exclusively from the training graph splits, thereby definitively ruling out potential data leakage regarding held-out test edges via triangle membership or coface counts.

5. The empirical evaluation must be standardized under a single, unified benchmark protocol with identical edge splits, negative sampling strategies, and hyperparameter tuning budgets, replacing the direct copying of literature baselines under mismatched setups, correcting the OGB evaluation metrics to match official dataset specific ranking protocols such as Hits@K rather than generalized ROC AUC, and properly aligning the loss functions with their respective regression or classification task formulations.